# IRIS: An Iterative and Integrated Framework for Real-Time Causal Discovery

## Abstract

Causal discovery is fundamental to scientific research, yet traditional statistical algorithms face significant challenges, including expensive data collection, redundant examination of known relations, and unrealistic assumptions. Additionally, while recent LLM-based methods excel at identifying commonly known causal relations, they fall short in uncovering novel relations. We introduce IRIS (**I**terative **R**etrieval and **I**ntegrated **S**ystem for Real-Time Causal Discovery), a novel framework that addresses these limitations. Starting with a set of initial variables, IRIS automatically retrieves relevant documents, extracts variable values, and organizes data for statistical algorithms in real-time. Our hybrid causal discovery method combines statistical algorithms and LLM-based methods to discover existing and novel causal relations. The missing variable proposal component identifies missing variables, and subsequently, IRIS expands the causal graphs by including both the initial and the newly suggested variables. Our approach offers a scalable and adaptable solution for causal discovery, enabling the exploration of causal relations from a set of initial variables without requiring pre-existing datasets.[1]

## 1 Introduction

A fundamental task in various disciplines of science, including biology, economics and healthcare, is to find underlying causal relations and make use of them Kuhn (1962). Although interventional experiments are ideal for discovering causality, they are often impractical or impossible due to ethical, financial, or logistical constraints. Consequently, researchers have developed methods to infer causal relations from purely observational data Pearl (2009); Spirtes et al. (2000).

Both statistical and large language model (LLM) -based causal discovery algorithms face several significant challenges that limit their applicability and effectiveness in real-world scenarios. First, statistical algorithms typically require extensive sample collection, a process that can be both time-consuming and expensive. As a result, many studies rely on synthetic data, potentially limiting the generalizability of their findings Dong et al. (2023); Gasse et al. (2021); Korb & Nicholson (2010); Binder et al. (1997). Second, statistical algorithms often redundantly examine well-established causal relations, leading to inefficient use of computational resources, especially given the high complexity of many causal discovery algorithms Zhang et al. (2011); Glymour et al. (2019). Third, while LLM-based methods may identify well-established causal relations, they struggle to uncover novel causal relations not previously documented Feng et al. (2024); Zečević et al. (2023). Lastly, most statistical algorithms operate under assumptions that rarely hold in real-world scenarios, such as the *causal sufficiency* assumption (*i.e.,* the absence of unobservable variables in the causal graph) and *acyclicity* assumption (*i.e.,* the absence of cycles in the causal graph) Pearl (2009); Neal (2020).

To address these limitations, we propose a framework, called **I**terative **R**etrieval and **I**ntegrated **S**ystem (IRIS) for real-time causal discovery. IRIS begins with a set of initial variables and employs an automated process to collect and analyze unstructured text data in real-time, eliminating the need for pre-existing sample collection. This data is then transformed into structured format suitable for statistical causal discovery methods. Our framework utilizes a *hybrid* approach to causal discovery, merging statistical methods with LLM-based causal relation extraction and verification techniques. This hybrid strategy allows us to leverage existing knowledge while simultaneously uncovering novel causal relations. Specifically, this hybrid approach allows cycles in causal graphs,

---

[1]Our code and data are available at `https://anonymous.4open.science/r/iris-7378`

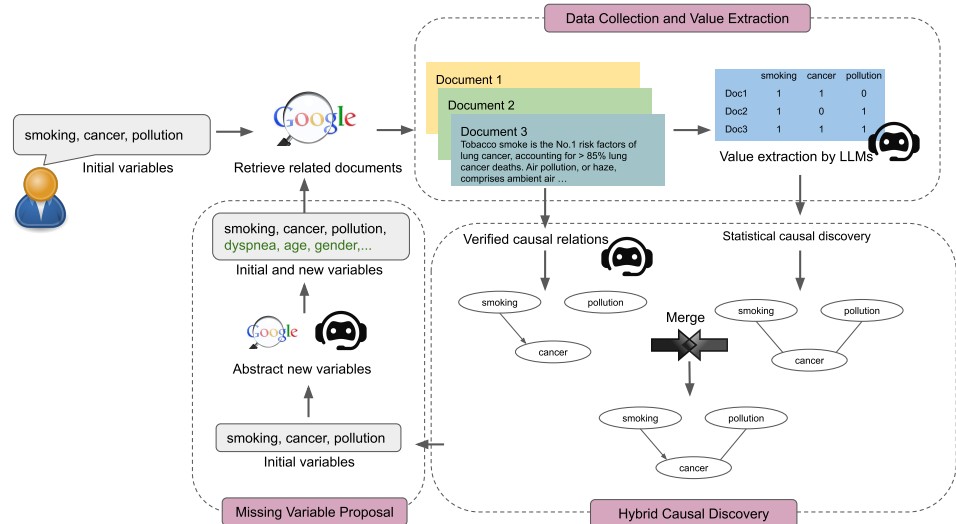

Figure 1: Illustration of our framework.

thereby discarding the *acyclicity* assumption. Additionally, we introduce a missing variable proposal component that identifies unobserved variables that may have causal associations with the initial variables. This component allows us to relax the *causal sufficiency* assumption. Then we can conduct an iterative process where the expanded variables are used as input, to apply to our framework again, resulting in expanded causal graphs.

Our experimental results demonstrate that IRIS significantly surpasses strong baselines across all datasets, achieving an average F1 score improvement of 0.14 and a reduction of 0.14 in the average NHD ratio, as detailed in Section 4. Evaluations of individual components reveal that each outperforms its corresponding baselines. Specifically, the value extraction assessment shows that IRIS with GPT-4o exceeds the strong baselines, which also utilizing GPT-4o (refer to Section 5). In terms of causal discovery, our hybrid method consistently outperforms both traditional statistical algorithms and LLM-based approaches, as outlined in Section 6. Lastly, the evaluation of our missing variable proposal component, discussed in Section 7, indicates that our method is more effective at correctly identifying missing variables compared to prompt-based baselines.

Our primary contributions are as follows: 1)We introduce a real-time sample collection and value extraction component that significantly reduces the manual labor required for data gathering in causal discovery. 2) We propose a hybrid causal discovery method that leverages existing causal knowledge through relation extraction and uncovers novel causal relations using statistical algorithms. Importantly, our method permits cycles in causal graphs, thus relaxing the *acyclicity* assumption. 3) We develop a missing variable proposal component that identifies unobserved variables that may have causal relations with the initial variables, facilitating the relaxation of the *causal sufficiency* assumption. 4) Our experimental evaluations demonstrate that IRIS consistently outperforms its baselines, with each component of IRIS also surpassing corresponding baseline methods.

## 2 BACKGROUND

Causal discovery focuses on uncovering causal structures within a set of variables. Given a pair of variables $(X, Y)$, the objective is to determine whether $X \leftarrow Y$, $Y \leftarrow X$, or no causal influence between them, where $\leftarrow$ denotes causal direction. A key distinction between causal discovery and relation extraction in NLP is that causal discovery reveals previously unknown causal relations, whereas relation extraction focuses on transforming relations in free text into structured relational tuples.

Although randomized controlled trials and A/B testing are the gold standard for causal discovery Fisher (1935), these experimental approaches are often impractical due to ethical or financial limitations. Thus, researchers turn to alternative methods that rely solely on statistical analysis of observational data to infer causal structures.

Statistical approaches to causal discovery can be broadly classified into two categories: constraint-based methods, exemplified by Peter and Clark (PC) Spirtes et al. (2000) and inductive causation (IC) Pearl (2009), and score-based methods Heckerman et al. (1995); Chickering (2002); Koivisto & Sood (2004); Mooij et al. (2016). These methodologies employ statistical measures derived from observational data to construct causal graphs. However, these approaches face significant challenges. Firstly, they require extensive data collection, which can be resource-intensive and time-consuming. Secondly, they are subject to theoretical limitations. These statistical methods cannot guarantee the precise identification of ground-truth causal graphs. Instead, they typically produce an equivalence class of true causal graphs Spirtes et al. (2000); Pearl (2009).

Furthermore, many causal discovery algorithms, such as PC and Greedy Equivalence Search (GES), operate under assumptions that can limit their applicability to real-world scenarios. *Causal sufficiency* assumption posits that all variables are observed and included, neglecting the potential unobserved variables Neal (2020). Some algorithms, such as Tetrad condition-based Silva et al. (2006); Kummerfeld & Ramsey (2016) and high-order moments-based approaches Adams et al. (2021); Chen et al. (2022) focus on uncover specific types of unobserved variables, such as latent confounders (i.e., common causes). However, our work aims to identify more general unobserved variables, including confounders, mediators, cause or effect of observed variables. *Acyclicity* assumption states that causal graphs contain no cycles, which allows causal discovery to align with Bayesian network and simplifies mathematical challenge. However, this assumption often contradicts real-world phenomena. Many causal processes are known to contain feedback loops, such as the poverty cycle: poverty → limited access to education → low-paying jobs → poverty, Banerjee & Duflo (2012); De Weiss & Sirkin (2010) and the predator-prey cycle: increase in predator population → decrease in prey population → decrease in predator population Schmitz (2017); Abrams (2001). In contrast to prior work, our proposed causal discovery method relaxes these assumptions. By allowing for the inclusion of unobserved variables and permitting cycles within causal graphs, our approach aims to model causal relations more closely aligned with real-world scenarios.

The advent of LLMs provides new opportunities to address causal discovery Kıcıman et al. (2023); Zečević et al. (2023); Long et al. (2022). In this approach, LLMs are prompted to determine the causal direction between a given pair of variable names. However, the reliability of such methods is under scrutiny. Zečević et al. (2023) argue that LLMs may function as *"causal parrots"*, which depend on *memorization* to recall the causal relations present in their training data rather than genuine causal reasoning. This raises concerns about the models' ability to *generalize* and identify causal relations that are rare or absent in pre-training data. Feng et al. (2024) presents empirical evidence to support this argument, suggesting that while LLMs may excel at reproducing frequently presented causal relations, they may struggle to uncover novel causal relations.

In contrast to approaches that directly employ LLMs for causal discovery, Liu et al. (2024) utilize LLMs as variable proposers to abstract causal variables and extract their values from collected documents, subsequently applying statistical methods to uncover causal relations among these variables. Our work diverges from this approach by taking a set of initial variables as input and employing an automated process to collect relevant documents. Following variable value extraction, our methodology implements a hybrid causal discovery approach, integrating both statistical methods and relation extraction methods. Furthermore, our approach is capable of proposing additional variables that exhibit causal relations with the initial set, thereby enabling an iterative process of data collection and causal discovery of a expanded set of variables. This iterative methodology allows for a more comprehensive exploration of the causal relations surrounding the variables of interest.

## 3 METHODOLOGY

In this work, we introduce a novel causal discovery framework, IRIS. Our method differs from prior causal discovery algorithms in three key aspects. First, IRIS does not rely on pre-existing observational data; instead, it automatically collects and extracts observational data related to the initial variables. Second, our approach does not require the *acyclicity* assumption; we allow cycles in the causal graph. Third, we do not assume *causal sufficiency*; our missing variable proposal component is designed to suggest new variables that may have causal relations with initial variables. IRIS consists of three principal components: Data Collection and Value Extraction, Hybrid Causal Discovery, and Missing Variable Proposal. We detail each component in the following sections.

## 3.1 Problem Definition

IRIS begins by taking a set of initial variables, denoted as $\mathbb{Z} = (z_1, z_2, ..., z_N)$, where $z_i$ represents one variable. Our method automatically collects a set of unstructured textual data $\mathbb{D}$ relevant to these initial variables. From $\mathbb{D}$, our method extracts the values of these variables to assemble structured data $\mathbb{X}$, which serves as the input for statistical causal discovery algorithms. Then our missing variable proposal component suggests new variables related to initial variables, resulting in an expanded set of variables $\mathbb{Z}_m$. The final output of our method is an expanded causal graph $\mathcal{G} = (\mathbb{Z}_m, \mathbb{R})$, where $\mathbb{R} = (r_1, r_2, ..., r_l)$ represents the set of causal relations (edges in the graph).

## 3.2 Data Collection and Value Extraction

The initial phrase of IRIS comprises two main steps: retrieval of relevant documents and extraction of variable value. The detailed procedure is outlined in Algorithm 1 in Appendix A.4.

**Retrieval of Relevant Documents** We employ the official Google API [2] to retrieve documents related to the initial variables $\mathbb{Z}$. To maximize the relevance to all initial variables, we construct search queries using a stepwise removal approach: 1) Begin with queries containing all variable names (*e.g.,* "smoking" AND "cancer" AND "pollution"). 2) Progressively remove one variable at a time (*e.g.,* "smoking" AND "cancer"). 3) Conclude with single-variable queries (*e.g.,* "smoking"). We also incorporate synonyms of variable names to enhance coverage (*e.g.,* "smoker" and "tobacco" for "smoking"). For each query, we retrieve the top-k documents to ensure data diversity. To ensure relevance to most variables, we require queries containing more variables yield more documents. The retrieval process continues until a predefined threshold of the number of collected documents is reached. The resulting document set is denoted as $\mathbb{D} = (d_1, d_2, .., d_T)$, where $d_i$ represents a single document.

**Extraction of Variable Value** The next step involves extracting variable values from the collected documents $\mathbb{D}$ to construct structured data $\mathbb{X}$. We leverage LLMs for this task. Given an LLM $M$, we design a prompt $l$ that incorporates one document $d_i$ and the description of one variable $z_j$. The variable description includes its name and the meaning of each value. We guide the LLM to generate responses following multiple thinking steps, mimicking human expert reasoning, and provide the final answer in a specific format Lin et al. (2024). This generation process can be denoted as $o_{ij} = M(l(d_i, z_j))$, where $o_{ij}$ is LLM's response regarding the value of variable $z_j$ in document $d_i$. We then extract the exact value $v_{ij}$ from this formatted response $o_{ij}$. By iterating through all variables and documents, we populate the structured data $X$ where each column represents a variable and each row corresponds to a document. The prompt template for value extraction is presented in Table 9 in Appendix A.5.

## 3.3 Hybrid Causal Discovery

With the collected unstructured data $\mathbb{D}$ and structured data $\mathbb{X}$, we employ a hybrid approach to causal discovery, leveraging both statistical methods and relation extraction techniques. This hybrid strategy allow us to leverage existing causal relation and uncover novel causal relations.

**Statistical Causal Discovery** For structured data $\mathbb{X}$, we employ statistical causal discovery algorithms such as the PC algorithm Spirtes et al. (2000), GES Chickering (2003), and NOTEARS Zheng et al. (2018). These methods determine causal structures by analyzing statistical relations between variables. For instance, the PC algorithm performs conditional independence tests between variable pairs, progressively expanding the conditioning sets to determine the presence of causal relations. These algorithms process structured data $\mathbb{X}$, which contains variables and samples, and use predefined hyperparameters to produce a causal graph $\hat{\mathcal{G}}_s$ as the output.

**Causal Relation Extraction and Verification** We introduce a complementary method inspired by causal relation verification Si et al. (2024); Wadden et al. (2022). In this approach, we treat each potential causal relation as a claim (*e.g.,* "smoking causes lung cancer") and retrieve documents

---

[2]https://developers.google.com/custom-search/docs/overview

containing both the cause and effect terms (*e.g.,* "smoking" AND "lung cancer"). To ensure the trustworthiness of retrieved documents, we restrict the search domain to reputable academic repositories [3]. We then employ LLMs to assess whether each document supports or refutes or not relates with the causal relation using a carefully designed prompt (see Table 10 in Appendix A.5). If a majority of documents support the causal relation, we incorporate it into a causal graph $\hat{\mathcal{G}}_v$. Otherwise, this causal relation should not exist in $\hat{\mathcal{G}}_v$.

**Graph Merging**    The two branches of our hybrid method produce two causal graphs: $\hat{\mathcal{G}}_s$ from the statistical algorithms and $\hat{\mathcal{G}}_v$ from the causal relation verification process. To merge these into a final causal graph $\hat{\mathcal{G}}$, we post-process the causal graph $\hat{\mathcal{G}}_s$ by adding high-confidence verified causal relations from $\hat{\mathcal{G}}_v$ and removing relations that are strongly refuted by the verification process. This merging strategy is adopted for two main reasons: firstly, the structured data $\mathbb{X}$ used in the statistical analysis might contain noise introduced during the value extraction phase; secondly, causal relations that are widely supported or refuted by the majority of documents are considered well-established knowledge and are deemed trustworthy. The detailed process of our hybrid causal discovery method is outlined in Algorithm 2 in Appendix A.4.

### 3.4 MISSING VARIABLE PROPOSAL

In this step, our goal is to identify missing variables that are not included in the initial variables but may serve as confounders, mediators, causes, or effects of initial variables.

**Variable Abstraction**    We leverage LLMs to abstract missing variables from the retrieved documents $\mathbb{D}$. For each document, the LLMs is instructed to abstract variables that may potentially have causal relations with the initial variables. The instruction involves analyzing the content of each document, identifying variables that could influence or be influenced by the initial variables, and then providing the most reliable variable in a specified format. The prompt for variable abstraction is provided in Table 11 in Appendix A.5.

**Variable Selection**    To select the most promising variables from the abstracted variables, we employ a dual approach combining relation extraction and statistical methods:

- *Verified Relation Extraction Approach*: We verify whether each abstracted variable has a confirmed causal relation with any initial variable using our causal relation extraction and verification method, as stated in Section 3.3. Variables that are supported by the majority of the documents are subsequently added to the expanded set of variables $\mathbb{Z}_m$.

- *Statistical Approach*: We compute the Pointwise Mutual Information (PMI) between each new variable and the initial variables. PMI quantifies the dependence between two variables, with higher scores indicating a greater likelihood of potential causal association. The PMI between two variables $(z_i, z_j)$ is computed as:

$$PMI(z_i, z_j) = \log \frac{p(z_i, z_j)}{p(z_i)p(z_j)} = \log \frac{\frac{o(z_i, z_j)}{C}}{\frac{o(z_i)}{C} \frac{o(z_j)}{C}} = \log \frac{o(z_i, z_j)}{o(z_i)o(z_j)} + \log C \quad (1)$$

where $o(z_i, z_j)$ denotes the number of documents where $(z_i, z_j)$ co-occur, and $o(z_i)$ represents the number of documents where $(z_i)$ occurs. These occurrence counts are obtained through Google search API. $C$ is a constant value that represents the total number of retrievable documents. Thus, $\log C$ can be ignored.

For each abstracted variable, we compute its PMI score with each initial variable. The top k abstracted variables with the highest aggregate PMI scores across all initial variables are appended to $\mathbb{Z}_m$. The detailed process of our missing variable proposal method is outlined in Algorithm 3 in Appendix A.4.

---

[3]Our search is limited to the following academic website domains: jstor.org, springer.com, ieee.org, ncbi.nlm.nih.gov, sciencedirect.com, scholar.google.com, arxiv.org.

With the newly proposed variables $\mathbb{Z}_m$, we can iterate the data collection, value extraction, and causal discovery processes to generate an expanded causal graph $\mathcal{G} = (\mathbb{Z}_m, \mathbb{R})$ that incorporates these additional variables and new discovered causal relations.

# 4 EVALUATION OF EXPANDED CAUSAL GRAPHS

## 4.1 EXPERIMENTAL SETUP

We assess the complete pipeline of IRIS, which includes causal discovery of initial variables, missing variables proposal, and subsequent causal discovery incorporating both initial and proposed variables. We then evaluate the quality of the resulting expanded causal graphs.

**Datasets.** The initial variables are from five datasets: Cancer Korb & Nicholson (2010), Respiratory Disease, Diabetes, Obesity Hernán et al. (2004); Long et al. (2022), and Alzheimer's Disease Neuroimaging Initiative (ADNI) Shen et al. (2020).

**Our Method and Baselines.** For our framework and baseline, we employs GPT-4o as the LLM component, a choice supported by its superior performance across value extraction, causal discovery, and missing variable proposal tasks (see Sections 5, 6, and 7). For the statistical causal discovery algorithms in our method, we utilize the Greedy Equivalence Search (GES) algorithm. This selection is based on GES achieving the highest average F1 score and Normalized Hamming Distance (NHD) ratio across all five datasets, as demonstrated in Section 6. To provide a comprehensive evaluation, we introduce a baseline method, coined "Prompt", which relies solely on carefully crafted prompts (see Table 12 in Appendix A.5) with LLM to determine causal relations among expanded variables proposed by our missing variable proposal component.

**Evaluation.** To establish ground-truth expanded causal graphs, we engage a panel of domain experts. Three knowledgeable annotators independently assess each graph, with edges included in the final ground-truth when at least two annotators agree. The inter-annotator agreement, calculated using Krippendorff's alpha, is 0.88, indicating a high level of agreement among annotators Krippendorff (2011). The detailed annotation instruction is in Table 13 in Appendix A.7. Following Kıcıman et al. (2023); Feng et al. (2024), we evaluate the results of causal discovery using precision, recall, F1 score, and the Ratio of Normalized Hamming Distance (NHD) to baseline NHD. The ratio is defined as ratio $= \frac{\text{NHD}}{\text{baseline NHD}}$, where the baseline NHD is derived from the worst-performing causal graph that has the same number of edges as the predicted graph. A lower ratio signifies a more accurate predicted causal graph.

## 4.2 EXPERIMENTAL RESULTS AND ANALYSIS

| Dataset | Method | Precision | Recall | F1↑ | # of predicted edges | NHD Ratio↓ |
|---|---|---|---|---|---|---|
| Cancer | Prompt | 0.64 | 0.32 | 0.43 | 14 | 0.57 |
| | IRIS | 0.89 | 0.57 | **0.7** | 18 | **0.3** |
| Respiratory | Prompt | 0.67 | 0.36 | 0.47 | 12 | 0.53 |
| | IRIS | 0.67 | 0.55 | **0.6** | 18 | **0.4** |
| Diabetes | Prompt | 0.70 | 0.46 | 0.56 | 17 | 0.45 |
| | IRIS | 0.76 | 0.5 | **0.6** | 17 | **0.39** |
| Obesity | Prompt | 0.57 | 0.33 | 0.42 | 14 | 0.58 |
| | IRIS | 0.67 | 0.58 | **0.62** | 21 | **0.38** |
| ADNI | Prompt | 0.47 | 0.29 | 0.36 | 17 | 0.64 |
| | IRIS | 0.5 | 0.36 | **0.42** | 20 | **0.58** |

Table 1: Evaluation results of expanded causal graphs.

Table 1 presents the evaluation results for the expanded causal graphs. IRIS consistently outperforms the Prompt baseline across all datasets, achieving higher F1 scores and lower NHD ratios. The average F1 score improvement is 0.14. Similarly, the average NHD ratio decreased by 0.14. ADNI exhibits the lowest overall performance for both methods, though IRIS still shows improvement over the baseline. This may reflect the inherent complexity of Alzheimer's disease causal relations. IRIS predicts more edges than the baseline (averaging 18.8 vs. 14.8 edges), which ensures a higher recall than the baseline (averaging 0.51 vs. 0.35). This indicates that our method's hybrid causal

discovery can capture more complex causal structures effectively. The expanded causal graphs for each dataset are illustrated in Figures 3, 4, 5, 6, 7 in Appendix A.8. These results demonstrate that IRIS can reliably expand and discover causal relations.

## 5 EVALUATION OF VALUE EXTRACTION

### 5.1 EXPERIMENTAL SETUP

**Datasets.** We evaluate the value extraction component of our method using two table-to-text datasets: AppleGastronome and Neuropathic Liu et al. (2024). These datasets are particularly suitable for our task as they provide tabular data where columns represent variables and rows represent samples. Each row is associated with a corresponding textual description. The datasets are structured as follows: AppleGastronome contains 7 variables and 100 samples. Variable values are -1, 0, or 1. Neuropathic contains 7 variables and 100 samples. Variable values are 0 or 1.

**LLMs and Baselines.** We utilize state-of-the-art LLMs for our method: Llama-3.1-8b-Instruct Meta (2024), GPT-3.5-turbo OpenAI (2022), GPT-4o OpenAI (2024). Additionally, we compare our method with COAT, which also utilizes an LLM to extract values of factors from documents Liu et al. (2024). To ensure a fair comparison, we use GPT-4o in both our method and the COAT implementation.

**Metrics.** Given that variable values are categorical, we frame the value extraction task as a classification problem, predicting the value of a variable in a given document. Consequently, we employ standard classification metrics: precision, recall, and F1.

### 5.2 EXPERIMENTAL RESULTS AND ANALYSIS

Table 2 presents the evaluation results of our value extraction method across different LLMs on the AppleGastronome and Neuropathic datasets. Our method's superior performance with GPT-4o, compared to COAT using the same LLM, indicates that our approach is more effective than COAT under identical LLM. In both datasets, we observe a consistent trend of improvement from Llama-3.1-8b-Instruct to GPT-3.5, and further to GPT-4o when using our method. This progression aligns with the general understanding that more advanced LLMs tend to perform better on complex tasks. Overall, the models perform better on the Neuropathic dataset compared to AppleGastronome. This could be attributed to the simpler binary values of the Neuropathic dataset (values 0 or 1) compared to the ternary values in AppleGastronome (-1, 0, 1). The additional complexity in AppleGastronome might introduce more opportunities for misclassification. The high performance of GPT-4o suggests that it could be highly effective for value extraction in our framework.

| AppleGastronome | | | |
|---|---|---|---|
| | P | R | F1 |
| COAT - GPT-4o | 0.74 | 0.76 | 0.75 |
| IRIS- Llama | 0.71 | 0.72 | 0.71 |
| IRIS- GPT-3.5 | 0.75 | 0.77 | 0.76 |
| IRIS- GPT-4o | **0.79** | **0.82** | **0.79** |
| Neuropathic | | | |
| COAT - GPT-4o | 0.72 | 0.80 | 0.79 |
| IRIS- Llama | **0.76** | 0.82 | 0.79 |
| IRIS- GPT-3.5 | 0.71 | 0.89 | 0.79 |
| IRIS- GPT-4o | 0.73 | **1.0** | **0.84** |

Table 2: Result of evaluation of value extraction. Llama represents Llama-3.1-8b-instruct

## 6 EVALUATION OF CAUSAL DISCOVERY

### 6.1 EXPERIMENTAL SETUP

**Datasets.** We apply our hybrid causal discovery component to five datasets: Cancer, Respiratory Disease, Diabetes, Obesity, and ADNI. These causal graphs are annotated by domain experts. The ground-truth causal graphs are presented in Figure 2 in Appendix A.6.

**Baselines.** We compare our method against several baselines: 1) Pairwise-LLM constructs queries for each pair of variables, using LLMs to determine causal relations. The computational complexity of this method is $O(n^2)$ Feng et al. (2024). 2) BFS-LLM employs a breadth-first search approach with LLMs, achieving linear computational complexity Jiralerspong et al. (2024). 3) COAT utilizes

LLM to extract values from documents, then applies the PC algorithm for causal discovery Liu et al. (2024). In our hybrid approach, for statistical algorithms, we utilize PC Spirtes et al. (2000), GES Chickering (2003), and NOTEARS Zheng et al. (2018). Among the three statistical methods (GES, NOTEARS, PC), we select the one that demonstrates the best performance for hybrid causal discovery. Based on our value extraction results (see Table 2), we use GPT-4o, which demonstrated the best performance, as the LLM for both our method and the baseline approaches. To illustrate how different LLMs affect the performance of our method, we employ the Llama-3.1-8b-instruct model as a counterpart.

**Metrics.** We evaluate the quality of causal graphs using precision, recall, F1, and NHD ratio as detailed in Section 4.

| Cancer (4 edges, 5 nodes) | | | | | |
|---|---|---|---|---|---|
| Method | Precision | Recall | **F1↑** | # of predicted edges | **NHD Ratio↓** |
| Pairwise-LLM | 0.75 | 0.75 | 0.75 | 4 | 0.25 |
| BFS-LLM | 0.6 | 0.75 | 0.67 | 5 | 0.33 |
| COAT | 0.13 | 0.25 | 0.17 | 8 | 0.83 |
| IRIS- GES | 0.25 | 0.5 | 0.33 | 8 | 0.67 |
| IRIS- NOTEARS | 1.0 | 0.25 | 0.4 | 1 | 0.6 |
| IRIS- PC | 0.14 | 0.25 | 0.18 | 7 | 0.82 |
| IRIS- VCR | 1.0 | 0.75 | **0.86** | 3 | **0.14** |
| IRIS (Llama) - NOTEARS+VCR | 0.375 | 0.75 | 0.5 | 8 | 0.5 |
| IRIS- NOTEARS+VCR | 1.0 | 0.75 | **0.86** | 3 | **0.14** |

Table 3: Evaluation results of causal discovery on cancer graph. VCR refers to verified causal relations that are extracted from and validated by relevant academic documents. "Llama" refers to the use of the Llama-3.1-8b-instruct model as a substitute for GPT-4o in our method.

| Respiratory Disease (5 edges, 4 nodes) | | | | | |
|---|---|---|---|---|---|
| Method | Precision | Recall | **F1↑** | # of predicted edges | **NHD Ratio↓** |
| Pairwise-LLM | 1.0 | 0.6 | 0.75 | 3 | 0.25 |
| BFS-LLM | 0.67 | 0.4 | 0.5 | 3 | 0.5 |
| COAT | 1.0 | 0.8 | 0.89 | 4 | 0.11 |
| IRIS- GES | 1.0 | 0.8 | 0.89 | 4 | 0.11 |
| IRIS- NOTEARS | 1.0 | 0.2 | 0.33 | 1 | 0.67 |
| IRIS- PC | 0.83 | 1.0 | 0.91 | 6 | 0.09 |
| IRIS- VCR | 1.0 | 0.8 | 0.89 | 4 | 0.11 |
| IRIS (Llama) - PC+VCR | 1.0 | 0.8 | 0.89 | 4 | 0.11 |
| IRIS- PC+VCR | 0.83 | 1.0 | **0.91** | 6 | **0.09** |

Table 4: Evaluation results of causal discovery on respiratory disease graph.

| Diabetes (5 edges, 4 nodes) | | | | | |
|---|---|---|---|---|---|
| Method | Precision | Recall | **F1↑** | # of predicted edges | **NHD Ratio↓** |
| Pairwise-LLM | 0.67 | 0.4 | 0.5 | 3 | 0.5 |
| BFS-LLM | 0.67 | 0.4 | 0.5 | 3 | 0.5 |
| COAT | 0.25 | 0.2 | 0.22 | 4 | 0.78 |
| IRIS- GES | 0.5 | 0.6 | 0.55 | 6 | 0.45 |
| IRIS- NOTEARS | 0 | 0 | 0 | 0 | 1.0 |
| IRIS- PC | 0.25 | 0.2 | 0.22 | 4 | 0.78 |
| IRIS- VCR | 1.0 | 0.2 | 0.33 | 1 | 0.67 |
| IRIS (Llama) - GES+VCR | 0.67 | 0.4 | 0.5 | 3 | 0.5 |
| IRIS- GES+VCR | 1.0 | 0.6 | **0.75** | 3 | **0.25** |

Table 5: Evaluation results of causal discovery on diabetes graph.

## 6.2 EXPERIMENTAL RESULTS AND ANALYSIS

The results of our causal discovery experiments across five datasets are presented in Table 3, 4, 5, 6, 7. Our hybrid method consistently outperforms baseline methods across all datasets. This underscores the effectiveness of combining statistical algorithms with extracted knowledge.

| Obesity (5 edges, 4 nodes) | | | | |
|---|---|---|---|---|
| | Precision | Recall | **F1**↑ | # of predicted edges | **NHD Ratio**↓ |
| Pairwise-LLM | 0.83 | 1.0 | 0.91 | 6 | 0.09 |
| BFS-LLM | 0.6 | 0.6 | 0.6 | 5 | 0.4 |
| COAT | 0.25 | 0.2 | 0.22 | 4 | 0.78 |
| IRIS-GES | 0.25 | 0.2 | 0.22 | 4 | 0.78 |
| IRIS- NOTEARS | 0 | 0 | 0 | 2 | 1.0 |
| IRIS- PC | 0.25 | 0.2 | 0.22 | 4 | 0.78 |
| IRIS- VCR | 1.0 | 1.0 | **1.0** | 5 | **0** |
| IRIS (Llama) - PC+VCR | 0.83 | 1.0 | 0.91 | 6 | 0.09 |
| IRIS- PC+VCR | 1.0 | 1.0 | **1.0** | 5 | **0** |

Table 6: Evaluation results of causal discovery on obesity graph.

| ADNI (7 edges, 8 nodes) | | | | |
|---|---|---|---|---|
| Method | Precision | Recall | **F1**↑ | # of predicted edges | **NHD Ratio**↓ |
| Pairwise-LLM | 0.5 | 0.14 | 0.22 | 2 | 0.78 |
| BFS-LLM | 0.33 | 0.14 | 0.2 | 3 | 0.8 |
| COAT | 0.11 | 0.14 | 0.13 | 9 | 0.87 |
| IRIS- GES | 0.08 | 0.14 | 0.11 | 12 | 0.89 |
| IRIS- NOTEARS | 0.33 | 0.14 | 0.2 | 3 | 0.8 |
| IRIS- PC | 0.11 | 0.14 | 0.13 | 9 | 0.87 |
| IRIS- VCR | 0.4 | 0.29 | 0.33 | 5 | 0.67 |
| IRIS (Llama) - NOTEARS+VCR | 0.08 | 0.14 | 0.11 | 12 | 0.89 |
| IRIS- NOTEARS+VCR | 0.38 | 0.43 | **0.4** | 8 | **0.6** |

Table 7: Evaluation results of causal discovery on ADNI graph.

We observe that the performance of individual statistical algorithms (GES, NOTEARS, PC) varied across datasets. PC excels in Respiratory Disease and Obesity. GES demonstrates optimal performance on Diabetes and Obesity. NOTEARS performs best on Cancer and ADNI but struggles significantly with Diabetes and Obesity, achieving a 0 F1 score and a 1 NHD ratio. This variation highlights the importance of selecting statistical algorithms based on the characteristics of the observational data, which presents a compelling area for further research. From our experiments, both GES and PC exhibit strong performances; however, GES consistently outperforms PC, with an average F1 score that is 0.09 points higher and an average NHD ratio that is 0.09 points lower. Given these results, GES is recommended as the primary choice when the suitability of the algorithm is uncertain. When comparing the performance of Llama-3.1-8b-instruct and GPT-4o in our method, GPT-4o consistently outperforms Llama-3.1-8b-instruct across all datasets, with a particularly significant gap observed in the ADNI dataset. We believe this discrepancy arises because ADNI involves specialized knowledge that is less commonly represented in the pre-training data of Llama-3.1-8b-instruct.

LLM-based methods (Pairwise-LLM and BFS-LLM) show competitive performance on simpler datasets. They perform well on the Cancer and Respiratory Disease datasets. However, their performance degrades on more complex datasets like ADNI. This suggests that while LLMs have potential in causal discovery, they may struggle with more complex causal relations, possibly due to the lower occurrence of such relations in their training data Feng et al. (2024). The COAT method yields results similar to IRIS- PC because both approaches extract values from documents and then perform causal discovery using the PC algorithm.

In conclusion, our experimental results consistently demonstrate that integrating the Verified Causal Relations (VCR) component with statistical algorithms significantly enhances causal discovery performance across datasets, thereby validating the effectiveness of our hybrid approach.

# 7 EVALUATION OF MISSING VARIABLE PROPOSAL

## 7.1 EXPERIMENTAL SETUP

**Datasets.** Evaluating the missing variable proposal component presents a unique challenge: the ground-truth missing variables are inherently unknown in real-world scenarios. To address this, we

simulate missing variables and assess our method's ability to identify them. We start with complete, ground-truth causal graphs and systematically remove variables to create incomplete graphs. We employ five causal graphs: Cancer, Respiratory Disease, Diabetes, Obesity, and ADNI. For each causal graph, we iteratively remove one variable at a time, creating multiple test cases per graph. We then apply our missing variable proposal component to these incomplete graphs, aiming to identify the removed variables.

**Our Method and Baselines.** For our missing variable proposal component, we employ GPT-4o and Llama-3.1-8b-instruct as the LLM. To ensure a thorough evaluation, we have introduced a baseline method that employs LLM to directly suggest new variables through a prompt-based approach.

**Metrics.** We evaluate the performance using a *success rate* metric, calculated as follows: 1) For each incomplete causal graph, we check if our method successfully proposes the removed variable in its final set of proposed variables $\mathbb{Z}_m$. 2) We count a "success" for each correctly proposed variable. 3) The success rate is computed as: Success Rate = Number of Successes/Total Number of Incomplete Graphs. For instance, in a causal graph with five variables, we create five different incomplete graphs by removing one variable at a time. If our method correctly proposes the removed variable in three of these five graphs, the success rate would be 0.6. For the statistical approach, we select the top-5 variables based on their PMI scores.

## 7.2 EXPERIMENTAL RESULTS AND ANALYSIS

The evaluation results of our Missing Variable Proposal (MVP) component are presented in Table 8. The MVP-GPT-4o method consistently outperforms other variants across all datasets. This demonstrates the effectiveness of combining VCR with statistical approach in identifying missing variables. Ablation studies indicate that both VCR and statistical approaches play a crucial role in enhancing the success rate of the MVP. The performance gap between MVP-GPT-4o and MVP-Llama indicates the superior capability of GPT-4o in understanding and reasoning about causal relations. The prompt-based baseline consistently underperforms compared to our framework, indicating that relying solely on the internal knowledge of LLMs is not reliable for proposing missing variables.

| | | Success rate | | | |
|---|---|---|---|---|---|
| Method | Cancer | Respiratory Disease | Diabetes | Obesity | ADNI |
| Prompt - GPT-4o | 0.4 | 0.25 | 0.5 | 0.25 | 0.25 |
| MVP - GPT-4o - NoVCR | 0.6 | 0.75 | 0.5 | 0.75 | 0.25 |
| MVP - GPT-4o - NoStats | 0.6 | 0.75 | 0.75 | 1.0 | 0.375 |
| MVP - Llama | 0.4 | 0.5 | 0.25 | 0.5 | 0.125 |
| MVP - GPT-4o | **0.8** | **0.75** | **1.0** | **1.0** | **0.5** |

Table 8: Evaluation results of the missing variable proposal (MVP) component. MVP-NoVCR excludes verified causal relation extraction; MVP-NoStats omits statistical approaches; MVP-Llama utilizes the Llama-3.1-8b-instruct model.

## 8 CONCLUSION

In this paper, we introduce IRIS, a novel framework that addresses several longstanding challenges in the field of causal discovery. By integrating automated data collection, hybrid causal discovery methods, and missing variable proposal components, IRIS significantly advances our ability to uncover causal relations in real-world scenarios. Our approach not only reduces the reliance on extensive manual data collection but also leverages existing knowledge while facilitating the discovery of novel causal relations. The ability to propose potentially missing variables allows for the development of comprehensive causal graphs. Our experimental results show that IRIS consistently outperforms existing baselines. Future work could aim to enhancing the scalability of IRIS for larger and more complex causal relations and to explore innovative methods for integrating causal relations extracted from texts with those identified through statistical algorithms.

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

# A APPENDIX

## A.1 RELATED WORK

**Causal Discovery** Causal discovery aims to uncover causal structures among variables, distinguishing itself from relation extraction in NLP by revealing novel causal relations rather than merely extracting known relations. While experimental approaches such as randomized controlled trials are gold standard methodsFisher (1935), practical limitations often necessitate statistical methods using observational data. These include constraint-based and score-based approaches Spirtes et al. (2000); Pearl (2009); Heckerman et al. (1995). However, statistical methods face challenges in data collection and theoretical limitations. Recent advancements in LLMs have introduced new possibilities for causal discovery without direct data access Kıcıman et al. (2023); Zečević et al. (2023); Long et al. (2022). However, concerns about LLMs functioning as "causal parrots" and their ability to generalize to novel relations have been raised Zečević et al. (2023); Feng et al. (2024). Alternative approaches, such as using LLMs for variable proposer and combining them with statistical methods Liu et al. (2024), have emerged. Our work builds upon these ideas, introducing an automated document collection process, a hybrid causal discovery method integrating statistical and relation extraction techniques, and a hybrid approach for new variable proposal.

**Relation Extraction** Relation extraction aims to transform unstructured textual relations into structured relation tuples of the form $< e_1, r, e_2 >$, where $e_1$ and $e_2$ represent entities and $r$ denotes the relation between them Yang et al. (2022); Dasgupta et al. (2018). While relation extraction can identify cause-effect relationships from documents, it fundamentally differs from causal discovery in that it relies on explicitly stated relations in texts, whereas causal discovery can uncover novel causal relationships from observational data even in the absence of explicit textual mentions. Nevertheless, relation extraction can serve as a complementary method for identifying commonly known causal relations in textual data. Several studies have focused on extracting causal relations from natural language texts Balashankar et al. (2019); Bui et al. (2010); Chang & Choi (2006). The methods for causality extraction can be divided into pattern-based and deep learning-based approaches. Pattern-based methods utilize predefined linguistic patterns to extract relevant text segments, which are then converted into tuples using hand-crafted algorithms Garcia (1997); Khoo et al. (2000). However, these methods often suffer from limited coverage of causal relations and require significant effort in pattern design. Deep learning-based methods employ neural networks to learn high-level abstract features and representations from sentences, framing relation extraction as a sequence-to-sequence task Zhao et al. (2023; 2024). While these approaches offer improved performance, they typically require large fine-tuning datasets and may not consistently produce structurally correct output tuples.

A notable limitation of many relation extraction systems is the lack of verification for extracted relations, potentially leading to the extraction of false or unreliable relations from untrustworthy sources Si et al. (2024); Wadhwa et al. (2023). Our work addresses this issue by adopting a novel approach: instead of directly extracting causal relations from documents, we pre-create textual mentions of causal relations (e.g., "smoking causes lung cancer") and employ LLMs to verify the veracity of these relations based on relevant documents. We consider a causal relation to hold if the majority of documents support its veracity, thereby enhancing the reliability of our extracted causal relations.

**Claim Verification** Claim verification aims to assess the veracity of claims based on relevant documents Bekoulis et al. (2021). This process typically encompasses several key components: claim detection, document retrieval, veracity prediction, and explanation generation. Research in this field often focuses on specific aspects of the verification pipeline. For instance, Panchendrarajan & Zubiaga (2024) and Li et al. (2024) concentrate on identifying check-worthy statements from large text corpora. Others, such as Wadden et al. (2022) and Mohr et al. (2022), prioritize veracity prediction, while Wang & Shu (2023) emphasize the importance of generating explanations for verification outcomes. The emergence of LLMs has significantly influenced the field, with numerous studies leveraging LLMs for claim verification through carefully crafted prompts Kim et al. (2024); Bazaga et al. (2024); Asai et al. (2024). Building on these advancements, one branch of our hybrid causal discovery approach reframes causal discovery as a causal relation verification task. We employ LLMs to assess the veracity of causal relations based on retrieved documents, subsequently incorporating verified relations into a causal graph. This methodology bridges the gap between tra-

ditional claim verification techniques and causal discovery, offering a novel approach to uncovering and validating causal relations.

## A.2 REPRODUCIBILITY STATEMENT

We release our code and scripts at `https://anonymous.4open.science/r/iris-7378`. Detailed descriptions of the algorithms used in each component of our framework can be found in Appendix A.4. We provide all prompts utilized throughout our framework in Appendix A.5. The ground-truth causal graphs employed in our evaluation experiments are outlined in Appendix A.6. Additionally, Appendix A.7 presents human annotation instruction and interface for the human annotation tasks involved in evaluating the expanded causal graphs. The annotated expanded causal graphs, alongside the predicted causal graphs, are documented in Appendix A.8.

## A.3 ETHICS STATEMENT

Our framework collects publicly available data that does not involve personal or sensitive information. However, uncovering causal relations from vast amounts of unstructured data can introduce biases inherent in the collected data and the LLMs used.

## A.4 ALGORITHMS

In this section, we provide detailed descriptions of the algorithms for each component of our method. The data collection and value extraction process is outlined in Algorithm 1. The hybrid causal discovery algorithm can be found in Algorithm 2. Finally, the algorithm for proposing missing variables is detailed in Algorithm 3.

---

**Algorithm 1** Document Collection and Value Extraction

---

**Require:** Initial Variables $\mathbb{Z}$, LLM $M$, threshold $T$, prompt $l$
  **Document Collection**
  $\mathbb{D} \leftarrow \emptyset$                                              ▷ Initialize an empty set for collected documents
  **while** $|\mathbb{D}| < T$ **do**
    $queries \leftarrow [(\mathbf{z}_1, \mathbf{z}_2, \ldots, \mathbf{z}_n), (\mathbf{z}_1, \mathbf{z}_2, \ldots, \mathbf{z}_{n-1}), \ldots, (\mathbf{z}_i)]$
                                                         ▷ queries considering all variables and their synonyms
    **for** each $q$ in $queries$ **do**
      $n \leftarrow 20 \times \text{len}(q)$                           ▷ Determine the number of URLs to collect
      $urls \leftarrow \text{google\_search}(q, n)$          ▷ Search with query $q$ and retrieve top-$n$ URLs
      **for** each $url$ in $urls$ **do**
        $D \leftarrow \text{extract text from } url$
        $\mathbb{D} \leftarrow \mathbb{D} \cup \{D\}$                       ▷ Add extracted text to the document set
      **end for**
    **end for**
  **end while**

  **Value Extraction**
  $\mathbf{V} \leftarrow$ Matrix of dimensions $T \times N$        ▷ Initialize a matrix with $T$ rows and $N$ columns
  **for** each $d_i$ in $\mathbb{D}$ **do**
    **for** each $\mathbf{z}_j$ in $\mathbb{Z}$ **do**
      $o_{ij} \leftarrow M(l(d_i, \mathbf{z}_j))$              ▷ Determine value of $\mathbf{z}_j$ in $d_i$ by LLM
      $v_{ij} \leftarrow \text{extract}(o_{ij})$                  ▷ Extract value from LLM output
      $\mathbf{V}[i][j] \leftarrow v_{ij}$           ▷ Store the value $v_{ij}$ in matrix $\mathbf{V}$ at position $(i, j)$
    **end for**
  **end for**
  **Output:** $\mathbb{D}, \mathbf{V}$

---

---

**Algorithm 2** Hybrid Causal Discovery

---

**Require:** Initial variables $\mathbb{Z}$, LLM $M$, structured data $\mathbb{X}$, prompt $l$, hyperparameters $\alpha, \beta$
    **Statistical Causal Discovery**
    $\hat{\mathcal{G}}_s \leftarrow$ causal_discovery_alg($\mathbb{X}$)         $\triangleright$ Apply causal discovery algorithms (e.g., PC algorithm)

    **Causal Relation Verification**
    $\hat{\mathcal{G}}_v \leftarrow$ causal graph with no edges
    remove_edges $\leftarrow \emptyset$
    **for** each $z_i$ in $\mathbb{Z}$ **do**
        **for** each $z_j$ in $\mathbb{Z}$ **do**
            **if** $z_i \neq z_j$ **then**
                $r \leftarrow$ "$z_i$ causes $z_j$"
                $veracity_r \leftarrow \emptyset$                 $\triangleright$ Initialize the veracity list for relation $r$
                **for** each $d$ in $\mathbb{D}_{z_i, z_j}$ **do**       $\triangleright \mathbb{D}_{z_i, z_j}$ denotes documents containing both $z_i$ and $z_j$
                    $ver_d \leftarrow M(l(r, d))$       $\triangleright$ Determine the veracity of $r$ based on document $d$
                    $veracity_r \leftarrow veracity_r \cup \{ver_d\}$
                **end for**
                **if** $veracity_r.count(True) > \alpha \times len(veracity_r)$ **then**
                    $\hat{\mathcal{G}}_v \leftarrow \hat{\mathcal{G}}_v \cup \{r\}$                 $\triangleright$ Add relation $r$ to the causal graph $\hat{\mathcal{G}}_v$
                **else if** $veracity_r.count(False) > \beta \times len(veracity_r)$ **then**
                    remove_edges $\leftarrow$ remove_edges $\cup \{r\}$
                **end if**
             **end if**
        **end for**
    **end for**

    **Merge $\hat{\mathcal{G}}_s$ and $\hat{\mathcal{G}}_v$**
    **for** each edge $r$ in $\hat{\mathcal{G}}_v$ **do**
        $\hat{\mathcal{G}}_s \leftarrow \hat{\mathcal{G}}_s \cup \{r\}$                         $\triangleright$ Add relation $r$ to $\hat{\mathcal{G}}_s$
    **end for**
    **for** each edge $r$ in remove_edges **do**
        $\hat{\mathcal{G}}_s \leftarrow \hat{\mathcal{G}}_s \setminus \{r\}$               $\triangleright$ Remove relation $r$ from $\hat{\mathcal{G}}_s$ if it exists
    **end for**
    $\hat{\mathcal{G}} \leftarrow \hat{\mathcal{G}}_s$                               $\triangleright$ The final merged causal graph
    **Output:** $\hat{\mathcal{G}}$

---

## A.5 PROMPTS

In this section, we show prompts we used in IRIS in Table 9, 10, 11. The prompt used in the "Prompt" baseline in evaluation of expanded causal graphs is shown in Table 12.

---

Given a document: {doc}

Please complete the below task.
We have a variable named '{var}'. The value of variable '{var}' is True or False.
True indicates that the existence of '{var}' can be inferred from the document, whereas False suggests that the existence of '{var}' cannot be inferred from this document.
Based on the document provided, what is the most appropriate value for '{var}' that can be inferred?
Please form the answer using the following format.
First, provide an introductory sentence that explains what information will be discussed.
Next, list generated answer in detail, ensuring clarity and precision.
Finally, conclude the final answer of the inferred value for '{var}' using the following template:
The value of '{var}' is ____.

---

Table 9: The prompt for value extraction, where doc indicates the content of a document, var refers to a variable name.

## A.6 GROUND-TRUTH CAUSAL GRAPHS

The ground-truth causal graphs for causal discovery can be found in Figure 2.

---

**Algorithm 3** Missing Variable Proposal

---

**Require:** Initial variables $\mathbb{Z}$, LLM $M$, collected documents $\mathbb{D}$, prompt $l$, hyperparameter $\alpha$
    **Step 1: Abstraction of Missing Variable Candidates**
    $\mathbb{Z}_c \leftarrow \emptyset$                                          $\triangleright$ Initialize the set of candidates
    **for** each document $d$ in $\mathbb{D}$ **do**
        $\mathrm{z} \leftarrow M(l(\mathbb{Z}, d))$                     $\triangleright$ Abstract a candidate variable from document $d$
        $\mathbb{Z}_c \leftarrow \mathbb{Z}_c \cup \{\mathrm{z}\}$
    **end for**

    **Step 2: Missing Variable Proposal Based on Verified Causal Relations**
    $\mathbb{Z}_m \leftarrow \emptyset$                             $\triangleright$ Initialize the set of missing variables
    **for** each variable $\mathrm{z}_i$ in $\mathbb{Z}_c$ **do**
        **for** each given variable $\mathrm{z}_j$ in $\mathbb{Z}$ **do**
            $r_1 \leftarrow$ "$\mathrm{z}_i$ causes $\mathrm{z}_j$"
            $veracity_{r_1} \leftarrow \emptyset$                   $\triangleright$ Initialize the veracity list for relation $r_1$
            **for** each document $d$ in $\mathbb{D}_{\mathrm{z}_i, \mathrm{z}_j}$ **do**     $\triangleright$ $\mathbb{D}_{\mathrm{z}_i, \mathrm{z}_j}$ denotes documents containing both $\mathrm{z}_i$ and $\mathrm{z}_j$
                $ver_d \leftarrow M(l(r_1, d))$            $\triangleright$ Determine the veracity of $r1$ based on document $d$
                $veracity_{r_1} \leftarrow veracity_{r_1} \cup \{ver_d\}$
            **end for**
            **if** $veracity_{r_1}.count(\text{True}) > \alpha \times veracity_{r_1}.count(\text{False})$ **then**
                $\mathbb{Z}_m \leftarrow \mathbb{Z}_m \cup \{\mathrm{z}_i\}$                $\triangleright$ Add $\mathrm{z}_i$ to the set of proposed variables
            **end if**
            $r_2 \leftarrow$ "$\mathrm{z}_j$ causes $\mathrm{z}_i$"             $\triangleright$ Repeat the process for the reverse causal relation
            $\ldots$
        **end for**
    **end for**

    **Step 3: Missing Variable Proposal Based on Statistical Methods**
    $\mathbb{S} \leftarrow \emptyset$                                     $\triangleright$ Initialize a set for PMI scores
    **for** each variable $\mathrm{z}_i$ in $\mathbb{Z}_c$ **do**
        $s_i \leftarrow \emptyset$
        **for** each given variable $\mathrm{z}_j$ in $\mathbb{Z}$ **do**
            $s_{ij} \leftarrow \text{PMI}(\mathrm{z}_i, \mathrm{z}_j)$                 $\triangleright$ Compute PMI of $(\mathrm{z}_i, \mathrm{z}_j)$ by Equation 1
            $s_i \leftarrow s_i \cup \{s_{ij}\}$
        **end for**
        $\mathbb{S} \leftarrow \mathbb{S} \cup \{\sum(s_i)\}$                      $\triangleright$ Aggregate the PMI scores for $\mathrm{z}_i$
    **end for**
    $\mathbb{Z}_m \leftarrow \mathbb{Z}_m \cup \text{top-k}(\mathbb{S}, \mathbb{Z}_c)$         $\triangleright$ Select the top-k variables based on their PMI scores
    **Output:** $\mathbb{Z}_m$                          $\triangleright$ Return the final set of proposed missing variables

---

---
Given a document: {doc}

Please complete the below task.
We have a claim: '{claim}'. We need to check the veracity of this claim. The value of veracity is True or False or Unknown.
True indicates that the given document supports this claim,
False indicates that the given document refutes the claim.
Unknown indicates that the given document has no relation to the claim.
Please form the answer with a logical reasoning chain according to the following format.
First, provide an introductory sentence that explains what information will be discussed.
Next, list the logical reasoning chain in detail, ensuring clarity and precision.
Finally, conclude the veracity of claim '{claim}' using the following template:
The veracity of claim '{claim}' is ___.

---

Table 10: The prompt for causal relation verification, where doc indicates the content of a document, claim refers to a causal relation (*e.g.,* smoking causes lung cancer).

Given a document: {doc}

Please complete the below task.
We have some given variables: '{observed_variables}'.
What are the high-level variables in the provided document that have causal relations to variables in the given variable set?
Please form the answer using the following format.
First, propose as many variables as possible that have causal relationships with the given variables, based on your understanding of the document.
Please ensure these proposed variables are different from the ones already provided.
Next, refine your list of candidate variables by reducing semantic overlap among them and shortening their names for clarity.
Finally, determine the most reliable variable candidate as the final answer using the template provided below:
The new abstracted variable is ____.

Table 11: The prompt for missing variable abstraction.

The task is to determine the cause-effect relation between two variables.
The variables are: {variable1} and {variable2}.
The answer should be {variable1} ->{variable2} or {variable1} <- {variable2} or no causal relation.
Let's provide a step-by-step process to analysis the relation between them,
then provide your final answer using the following format:
The final answer is ____.

Table 12: The prompt used in the baseline for evaluation of expanded causal graphs.

## A.7 CAUSAL RELATION ANNOTATION TASK

The detailed instructions for the causal relation annotation task are presented in Table 13. This table provides comprehensive guidance to annotators on how to identify and annotate causal relations among the given variables.

**Causal Relation Annotation Task**

**Task overview:**
Your task is to identify and annotate causal relations among a set of variables. A causal relation exists when one variable directly influences another.

**Instructions:**
1. Consider each pair of variables and determine if there is a direct causal relationship between them.
2. If you believe variable A causes variable B, indicate this as: A → B
3. Be cautious of confusing correlation with causation. Only mark a relationship if you believe there is a direct causal link.
4. Consider the direction of causality carefully. For example, "Obesity → Heart Failure" suggests obesity causes heart failure, not the other way around.
5. It's okay to have multiple causes for a single effect, or multiple effects from a single cause.
6. Not all variables will necessarily have causal relationships with others.
7. Use your best judgment based on available knowledge and logical reasoning.

**Examples:**
lifestyle -¿ obesity
heart defect -¿ cardiac output
genetic disorder -¿ heart defect

**Submission:**
Please submit your annotations as a list of causal relations in the format: Variable A -¿ Variable B
Thank you for your careful consideration of this task!

**Task 1: Cancer**

**Variables:**
pollution
smoker
cancer
x-ray
dyspnoea
air quality
education
health issues
toxicity
chronic illness
covid-19
inflammation
respiratory issues
immunity
carcinogens
early detection

**Causal Relations:**
...

Table 13: Instructions and interface of causal relation annotation task.

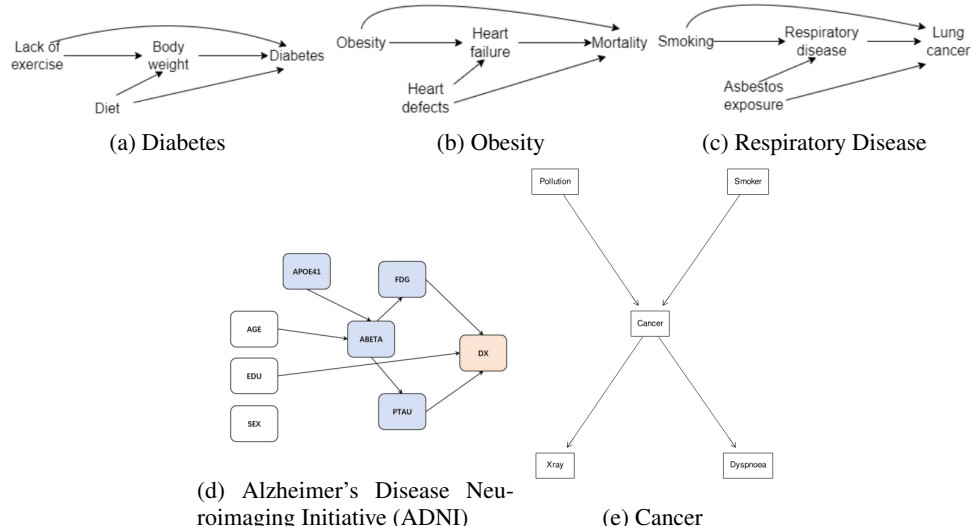

(a) Diabetes      (b) Obesity      (c) Respiratory Disease

(d) Alzheimer's Disease Neuroimaging Initiative (ADNI)

(e) Cancer

Figure 2: The ground-truth causal graphs from original sources Hernán et al. (2004); Long et al. (2022); Shen et al. (2020); Korb & Nicholson (2010).

## A.8 EXPANDED CAUSAL GRAPHS

The expanded causal graphs are demonstrated in Figure 3, 4, 5, 6, 7.

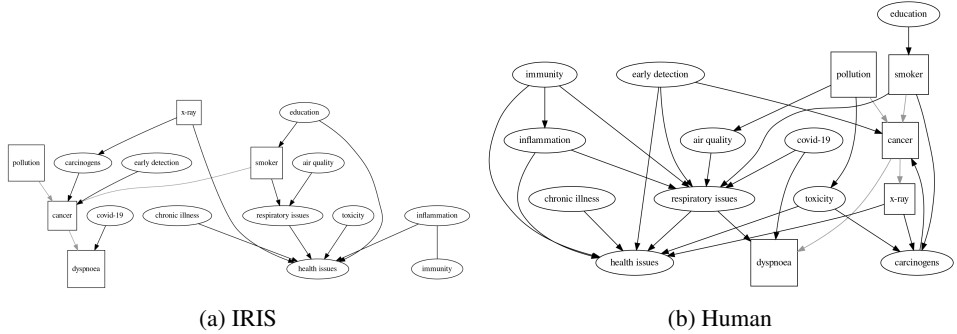

(a) IRIS      (b) Human

Figure 3: Illustration of expanded causal graphs for Cancer. Squared nodes represent initial variables, while round nodes denote new proposed variables.

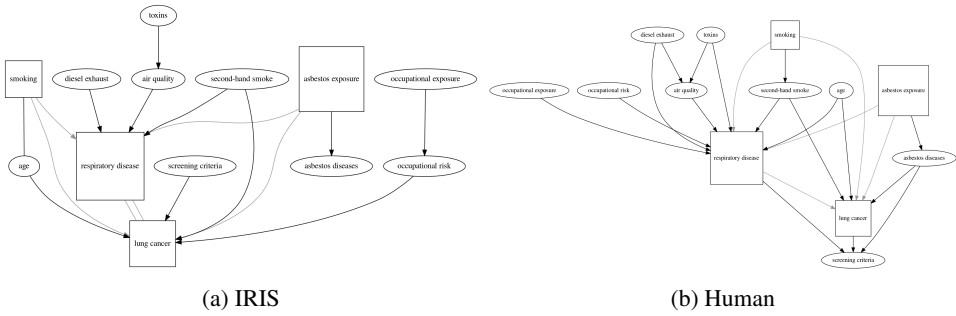

(a) IRIS      (b) Human

Figure 4: Illustration of expanded causal graphs for Respiratory Disease. Squared nodes represent initial variables, while round nodes denote new proposed variables.

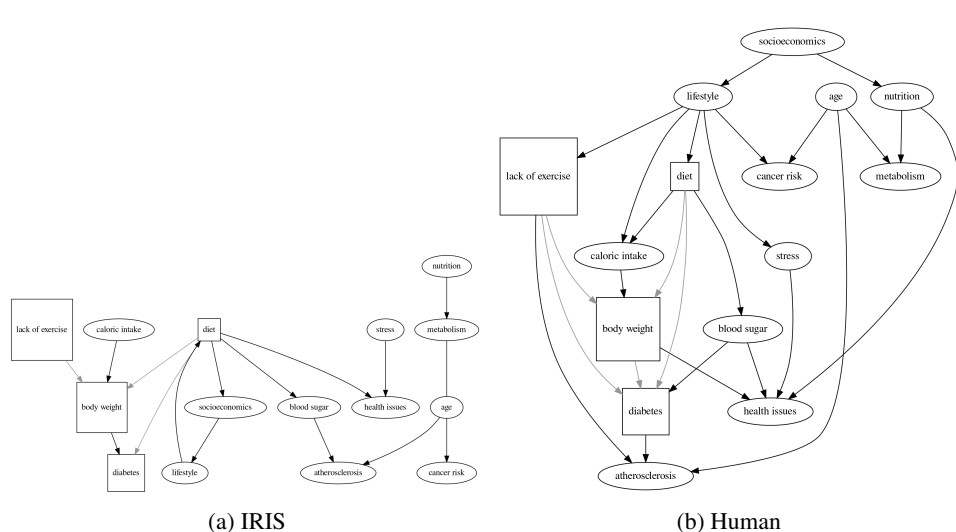

(a) IRIS  (b) Human

Figure 5: Illustration of expanded causal graphs for Diabetes. Squared nodes represent initial variables, while round nodes denote new proposed variables.

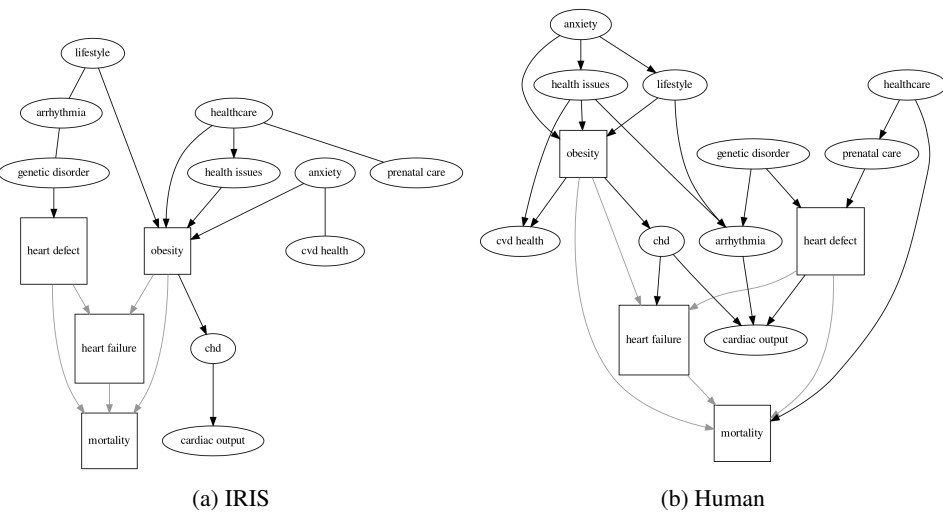

(a) IRIS  (b) Human

Figure 6: Illustration of expanded causal graphs for Obesity. Squared nodes represent initial variables, while round nodes denote new proposed variables.

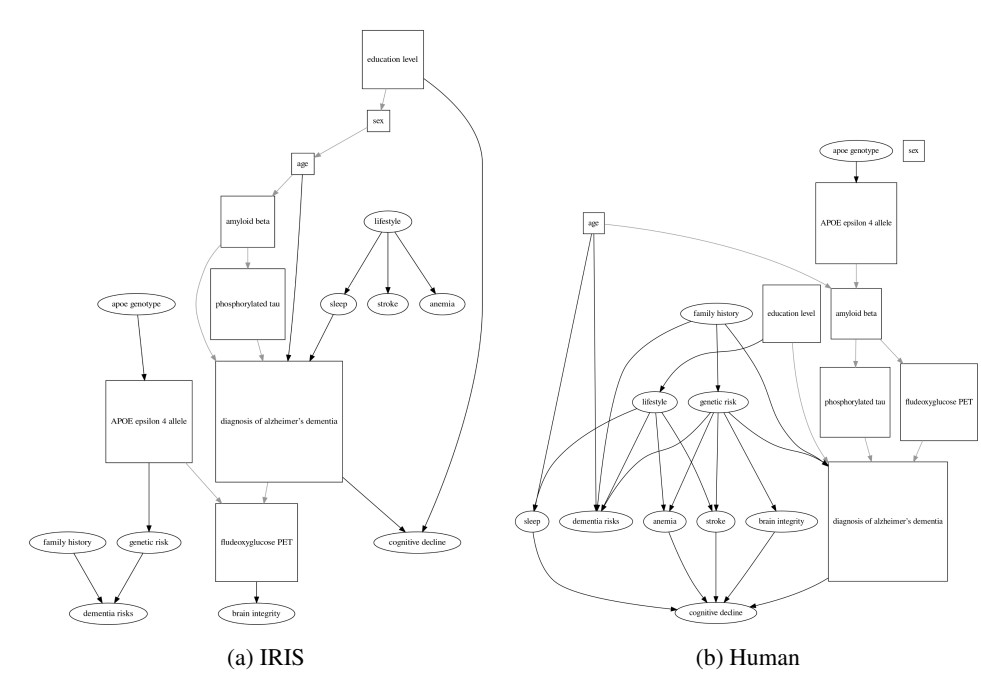

(a) IRIS                                        (b) Human

Figure 7: Illustration of expanded causal graphs for ADNI. Squared nodes represent initial variables, while round nodes denote new proposed variables.

