# OpenReview forum: "IRIS: An Iterative and Integrated Framework for Real-Time Causal Discovery"
_ICLR.cc/2025/Conference — ICLR 2025 Conference Withdrawn Submission_

### Official Review · Reviewer_CE1b · 2024-10-29

**Soundness:** 1
**Presentation:** 3
**Contribution:** 2
**Rating:** 3
**Confidence:** 4

**Summary:**

Authors propose a novel causal discovery technique called IRIS, an Iterative Retrieval and Integrated System for Real-Time Causal Discovery. This algorithm start from a set of variables and search the web to retrieve documents relevant to these keywords. It then proceeds to extract variable values (namely, the levels of a categorical variable) and extract keywords frequencies across documents. These quantities are used to construct a causal graph. From this initial graph, potential missing variable are identified and the process is repeated and the graph expanded. This approach relies on LLMs for both documents retrieval, value extraction and missing proposal. These steps are evaluated w.r.t. different benchmarks.

**Strengths:**

- An iterative approach for causal discovery allows to stop whenever a sufficiently stable solution is found, lowering the computational complexity.
- Automated data collection can be useful to bootstrap the causal discovery procedure.
- Missing variables proposal is an interesting approach to causal insufficiency that should be investigated more.

**Weaknesses:**

- The data collection and value extraction is performed by querying large databases (search engines) with some coarse queries based on iterative keywords reduction.
- In this paper, the "statistical causal discovery" is used to compute relationships based on terms-documents frequency. It has nothing to do with the literature cited by the authors that computes statistical properties based on actual data about the variables of interest.
- LLMs are trained on a large number of documents, we do not have any guarantees that they are not trained on the very same papers used to publish the datasets on which this method is evaluated. We cannot conclude anything about generalization and validity of the proposed approach.

**Questions:**

- It is unclear to me why acyclicity is a limitation. What does acyclicity means in terms of causal-effect relationships? What exactly is limited by making this assumption? Is this a limitation in general or in specific applied domains?
- Authors state that: "Many causal processes are known to contain feedback loops" . This is true only if we consider "processes", that is,  time. This should be made explicit. Is time modeled implicitly? Are the authors disregarding the difference between static and dynamic causal models?
- The existence of an equivalence class is given by specific parametric assumptions: there are probability distributions that are known to produce indentifiable causal graphs, i.e. DAGs. Furthermore, there are algorithms that produce DAGs, not CPDAGs, by design. Why are equivalence classes a problem if they are an essential part in describing the cause-effect relationships?
- Why the notation differs from the references (e.g. Pearl)? For instance: the usual notation is  " -> ", not " <- "; D, X, G = (V, E) is found in the references; X bold not defined; G hat s not defined; NHD should be better defined separately.
- How can we assure that the retrieved documents are relevant to our specific context? How can we assure that those are not fake? scholar.google.com, arxiv.org contains also non-peer reviewed papers, those are far from being "reputable academic repositories".
- Are we implicitly assuming that variables are categorical only?
- Can the authors provide a formal definition of "veracity" in the context of causal graphs?
- LLMs are trained on the very same papers that describe the datasets authors used during evaluation. How can we test a model that has been trained on the train set?

---

### Official Review · Reviewer_t6W4 · 2024-10-30

**Soundness:** 2
**Presentation:** 3
**Contribution:** 2
**Rating:** 3
**Confidence:** 4

**Summary:**

This paper introduces a framework for causal discovery that combines several methods, including Google searches for retrieving relevant documents, large language models (LLMs) for identifying known relationships, causal discovery techniques for uncovering causal links, and variable abstraction for detecting unobserved causes of initial variables.

**Strengths:**

•	The paper addresses an important problem, showing potential by integrating various existing techniques.

•	It includes extensive experimental evaluations of causal discovery using publicly available data, showcasing how the potential of the proposed framework.

**Weaknesses:**

I have concerns regarding the use of causal discovery methods, such as PC, GES, and NOTEARS, on the value-extraction dataset:

1.	Applicability of Causal Discovery Methods: Causal discovery methods are typically designed for observational data derived from naturally occurring or “real-world” events, often involving measured variables over time or across conditions, such as in longitudinal studies or surveys. However, the value extraction data here is aggregated from documents and lacks the context of natural experiments. The relationships identified among terms represent co-occurrence rather than causation. It seems unlikely that true causal relationships can be reliably uncovered from such summarized data.

2.	Assumptions of Causal Discovery Methods: Causal discovery methods require assumptions, such as acyclicity and causal sufficiency. When these assumptions are not met, strong justification is necessary to validate that the results remain causal. This paper appears to overlook these assumptions and does not provide evidence that the results are indeed causal.

3.	Experimental Results and Effectiveness: The experimental results indicate that causal discovery methods alone do not work effectively here; the highest-performing methods include Verified Causal Relations (VCR), suggesting that pre-verified causal knowledge is critical to success.

The experimental comparisons and conclusions raise further issues:

•	In Tables 3, 4, 5, 6, and 7, the methods reported vary inconsistently. For instance, some tables include IRIS (Llama)-PC + VCR and IRIS-PC+VCR, while others list IRIS (Llama)-GES + VCR and IRIS-GES+VCR, and yet others have IRIS (Llama)-NOTEARS + VCR and IRIS-NOTEARS + VCR. It is scientifically unsound to selectively report results across datasets in this manner.

•	There are a total of 10 IRIS variations and 3 comparison methods. Concluding that IRIS outperforms other methods based on one variation performing better than the comparison methods in a dataset introduces a potential for bias due to multiple comparisons.

**Questions:**

See Weaknesses.

---

### Official Review · Reviewer_RooZ · 2024-11-04

**Soundness:** 3
**Presentation:** 2
**Contribution:** 2
**Rating:** 3
**Confidence:** 3

**Summary:**

IRIS is an algorithm for using LLMs to construct causal graphs. IRIS starts with an initial set of variable Z,, and then uses LLMs to (i) collect relevant documents,(ii)  extract structured data about the original set of variables, (iii) apply causal discovery algorithms to the structured data to create a causal graph G,  (iv) extract more variables Znew related to the original set of variables Z, (v) extracts causal relations between Z and Znew, and (vi) merges the original graph G with the new causal relations. IRIS is tested on 5 different sets of initial variables taken from the epidemiology literature by comparing the graphs output by IRIS to causal graphs constructed by epidemiological experts. IRIS is found to outperform previous LLM algorithms.

**Strengths:**

The authors put a good deal of work into testing their algorithm. They took sets of variables from the epidemiology literature, and recruited 3 experts to compare gold standard causal graphs to compare their output to. Much of the exposition about how the algorithm worked was admirably clear, with a couple of glaring exceptions. They put effort into showing that each part of the hybrid algorithm contributed to the overall result. The results that they reported are limited, but given the cost of creating gold standard graphs, understandable. The results showed that their hybrid algorithm did outperform competitors.

**Weaknesses:**

The authors gave evidenc that IRIS improved the construction of causal graphs as compared to previous LLM algorithms, but it is unclear from the paper why that is.

The paper is confusing about how value extraction works, so I may have misunderstood it. If so, some of the following objections may be incorrect.  The structured data that comes out of value extraction is placed into a d x n table, call it Table 1, where the number of variables is n, and d is the number of documents. For a given variable and a given document, the LLM is asked:

    We have a variable named ’{var}’. The value of variable ’{var}’ is True or False.
    True indicates that the existence of ’{var}’ can be inferred from the document, whereas False suggests that
    the existence of ’{var}’ cannot be inferred from this document.
    Based on the document provided, what is the most appropriate value for ’{var}’ that can be inferred?

It is unclear whether the value extraction always forces the output to be binary. The prompt certainly makes it sound that way. However, the article applies value extraction to documents in which variables have 3 values, and makes it sound like it is supposed to extract the values of the variables. It is not clear why asking whether the existence of a variable can be inferred from a document would extract the value of a variable in that document. Or if a document said "Flouride does not cause cancer", whether it is possible to infer the existence of fluoride (or cancer) from that document. They report precision and recall for the case where the variables in the document had 3 values, so more information about what they meant by precision for this multiclass case would be helpful.

The value extraction was applied to the AppleGastrome and Neuropathic data sets. The AppleGastrome data set was created by using an LLM to create documents which were reviews of apples, where each document contained a description of the values of several features of an individual kind of apple, and whether the kind of apple was considered good. While IRIS outperformed another algoritm in extracting values of features in this case, this dataset seems very different from the kind of documents one would typically expect in epidemiology; those documents would not be descriptions of a single person and their attributes, so what the value extracted from such a paper should be is less clear. One thing that would help is to explain the AppleGastrome data looks like in more detail - I had to go to the original paper to get a sense of what the documents actually looked like.)

This raises the question of the use of the causal discovery algorithms on structured data. The structured data that is input into causal discovery algorithms in IRIS is very different from the usual kind of data the algorithms are designed for. In the usual setting, given 5 variables for example, a sample of units with values for those 5 variables would be measured, and the values of each of the variables for each of the units would be put into a table, call it Table 2.

Table 1 and Table 2 are very different. Table 2 can have arbitrary values for the values of its variables, while Table 1 can only have True or False (if I understand correctly). I don't see how recording a 10 for how many years someone has been smoking relates to a True of False answer to the question of whether it is possible to infer from some document that smoking  exists (or precisely what it would mean to infer the existence of smoking from a documentt.) Causal discovery algorithms are intended to work on tables of the kind described in Table 2. I don't see how they could be expected to work on tables of the kind described in Table 1, if only because a great deal of information is being thrown away when variables that can have many different values are being forced to be binary. They also depend on data being i.i.d., whereas the documents may well be dependent on each other, if for example one documents cites another. Also the documents are being selected on the basis of the terms they contain, so that could introduce selection bias. If the values extracted are always binary, much information is being thrown away, which could also affect the causal discovery algorithms.

In the examples of applications of IRIS, while IRIS did find other variables to add to the original set of variables, there were actually very few confounders among the new variables. That implies that the causal discovery algorithms could be relatively accurate in these examples, but one would not expect that to be true in general.

The way that information about new variables is integrated into the causal graph produced by the causal search algorithm is naive. For example, a statistical algorithm might orient edges as A -> B <- C because A and C are independent. But if background knowledge indicates that A and C are causally related, there is now a conflict between the background knowledge and the current orientations (because if A and C are dependent, the evidence that the orientations are A -> B <- C is gone.) One should not simply add the edge A -> C, but also remove the orientations of A ->B and B <- C. Also if A -> B is in the DAG, and a new variable X is added by IRIS along with the informaton A -> X -> B, it is not clear whether the edge A -> B should also be removed. (A is a direct cause of B relative to the set of variables without X, but may or may not be a direct cause of B relative to the set of variables with X.) How should this be handled?

**Questions:**

Are extracted values of variables always binary? If not, how were precision and recall calculated for multclass variables?
How does the prompt in A.5 relate to value extraction in the intuitive sense?
Given how the values of variables in IRIS are extracted, is there a justification of the use of causal discovery algorithms to structured data produced by IRIS?
Where did the list of variables presented to the human experts in A.7 come from?
Aren't the kind of documents in AppleGastrome very (description of a single kind of apple and its attributes) very different from what one would expect in e.g. an epidemiological document?
You state "This selection is based on GES achieving the highest average F1 score and Normalized Hamming Distance (NHD) ratio across all five datasets, as demonstrated in Section 6." I don't see any test of GES by itself in section 6, and GES does not have the highest F1 and Normalized Hamming Distance when it is part of a hybrid with IRIS. What did you mean here?

---

### Official Review · Reviewer_kb3v · 2024-11-05

**Soundness:** 2
**Presentation:** 3
**Contribution:** 1
**Rating:** 1
**Confidence:** 4

**Summary:**

The paper reports the design and evaluation of a system intended to assist human analysts with the task of constructing a causal model for a given domain by using text documents about the domain. The system (IRIS) takes a set of variables as input, retrieves relevant documents, proposes missing variables, extracts variable values, organizes data for structure learning algorithms, and learns the structure of those models.

**Strengths:**

The paper poses an interesting challenge in terms of using LLMs and large corpora of unstructured text to aid the process of constructing causal graphical models.

The authors attempt a quantitative evaluation.

**Weaknesses:**

The evaluation of IRIS focuses more on replicating human expectations than on accurately estimating causal effects. The “ground truth” for the causal graphical models constructed by IRIS are causal graphical models constructed by a set of human judges. This has several problems. First, this form of evaluation eliminates any possible mismatch between the formalism (graphical models) and actual causal effects in the world. Second, it merely checks (at best) whether the human judges reproduce what an LLM can extract from the expressions of causal judgments in textual training data. This avoids a key question: Whether the training data of LLMs expresses accurate causal knowledge. One of the key goals of causal inference is to provide a source of information about interventional effects that is separate from expert judgment. IRIS appears to reinforce those judgments, rather than check them.

The authors appear to show *that* the method improves accuracy, but not *why* it improves accuracy. Until a mechanism is clearly described and proven by detailed experiments, the results are unconvincing. On its face, it seems unlikely that sufficiently accurate quantitative data could be extracted from large text corpora and that such data would improve the construction of accurate causal graphical models.

The authors describe the construction of cyclic models, but they do not clearly define the semantics of those models. The introduction says: “Specifically, this hybrid approach allows cycles in causal graphs, thereby discarding the *acyclicity* assumption.” Accurate construction and use of cyclic models requires much more than simply *allowing* cycles. Instead, any use of cyclic models requires that the authors clearly define a semantics for cyclic graphs, as well as a corresponding semantics of inference in such graphs. They don’t do this. Thus, it is unclear what their cyclic models mean, in a formal sense.

A related problem is that the authors do not define a semantics of *temporal* causal models.  Such models pose additional challenges. One major challenge of doing this is to define a consistent time scale among the variables among which feedback occurs. In many real-world systems, some cycles of feedback occur in seconds and other cycles of feedback occur over days or months. These varying timescales have vexed many researchers who attempt to construct a useful semantics for cyclic causal models. The authors don’t even mention this or other similar challenges (e.g., hierarchical dependence, aggregation, etc.).

The authors incorrectly claim to relax the causal sufficiency assumption because of the effects of the IRIS component that proposes new variables. In reality, IRIS's ability to propose additional variables doesn’t relax the assumption. Instead, it makes it more likely that the assumption is met, provided that the added variables really are latent confounders. The authors should state this more clearly, and they should provide evidence in the paper that the added variables frequently correspond to latent confounders.

The authors do not provide evidence that the discovered variables satisfy necessary assumptions of causal graphical models. Variable definition is a major challenge of constructing such models. Specifically, variables in valid causal graphical model must meet a variety of assumptions, including the (quite challenging) assumption of modularity, also referred to as “autonomy” or “independence of causal mechanism”. Any process that purports to automatically discover useful variables would need to show that the discovered variables are modular (at a minimum). There is no evidence provided that the discovered variables meet this (or any other) standard.

The authors do not appear to effectively address key challenges of extracting structured data from text. Automatic extraction of structured data from large text corpora poses a variety of potential pitfalls, including variation in the definition of units and variables, heterogeneity in the conditions under which data are gathered (resulting in more, rather than less, latent confounding), and biased sampling (corresponding to conditioning on a collider and thus posing a new threat to internal validity). The authors don’t mention these threats to validity or how they address them.

It is unclear how the collection of data from documents is even possible. Essentially all traditional observational studies gather data from structured sources (e.g., national census data, surveys, etc.) rather than from text documents. For the example cited in the paper (“smoking” AND "cancer” AND “pollution”), it would seem unlikely that most text documents would contain specific instances from which useful data could be extracted (e.g., “John Smith smoked 15 cigarettes a day for 25 years and lived for most of that time in Pittsburgh which had 15 parts per million of pollution during that period”). The authors should make much clearer what sort of numeric data is appearing in documents and how it can be effectively extracted.

One writing issue interferes with effective reading of the paper. The authors provide citations without parentheses or other notation, making it difficult to separate citations from ordinary text on first reading. The authors should revise their citations to use parentheses (e.g., \citep).

**Questions:**

What evidence is provided in the paper about the specific mechanism by which information extraction improves the match between the CGMs constructed by human experts and those constructed with the help of IRIS?

What evidence is provided in the paper about the match between the judgments of human experts and actual ground truth causal effects?

What sort of structured data does IRIS collect and under what circumstances? Can you give some concrete examples of how data sets were improved by the IRIS process of finding documents and extracting numeric data?

---

### Note · Authors · 2024-12-15

I have read and agree with the venue's withdrawal policy on behalf of myself and my co-authors.